# Interactions of Tea-Derived Catechin Gallates with Bacterial Pathogens

**DOI:** 10.3390/molecules25081986

**Published:** 2020-04-23

**Authors:** Peter W. Taylor

**Affiliations:** School of Pharmacy, University College London, 29-39 Brunswick Square, London WC1N 1AX, UK; peter.taylor@ucl.ac.uk; Tel.: +44-(0)20-7753-5867

**Keywords:** catechin gallates, *Staphylococcus aureus*, β-lactam antibiotics, cytoplasmic membrane, penicillin-binding proteins, bacterial cell wall, divisome

## Abstract

Green tea-derived galloylated catechins have weak direct antibacterial activity against both Gram-positive and Gram-negative bacterial pathogens and are able to phenotypically transform, at moderate concentrations, methicillin-resistant *Staphylococcus aureus* (MRSA) clonal pathogens from full β-lactam resistance (minimum inhibitory concentration 256–512 mg/L) to complete susceptibility (~1 mg/L). Reversible conversion to susceptibility follows intercalation of these compounds into the bacterial cytoplasmic membrane, eliciting dispersal of the proteins associated with continued cell wall peptidoglycan synthesis in the presence of β-lactam antibiotics. The molecules penetrate deep within the hydrophobic core of the lipid palisade to force a reconfiguration of cytoplasmic membrane architecture. The catechin gallate-induced staphylococcal phenotype is complex, reflecting perturbation of an essential bacterial organelle, and includes prevention and inhibition of biofilm formation, disruption of secretion of virulence-related proteins, dissipation of halotolerance, cell wall thickening and cell aggregation and poor separation of daughter cells during cell division. These features are associated with the reduction of capacity of potential pathogens to cause lethal, difficult-to-treat infections and could, in combination with β-lactam agents that have lost therapeutic efficacy due to the emergence of antibiotic resistance, form the basis of a new approach to the treatment of staphylococcal infections.

## 1. Introduction

Antibiotics are among the most beneficial drugs ever introduced into clinical practice [1]. The dawn of the antibiotic era enabled previously untreatable infections to be swiftly brought under control and antibiotics contributed enormously to the rapid decline in morbidity and mortality from infectious diseases during the mid-twentieth century. In the first decades of the twenty-first century, the majority of nosocomial and community-acquired bacterial infections can be effectively treated with currently available antibiotics. However, the utility of our antibacterial armamentarium is being rapidly eroded by the emergence and spread, through horizontal gene transfer, of antibiotic resistant, multi-drug resistant (MDR) and pan-resistant bacterial pathogens brought about by Darwinian natural selection due to the overuse and misuse of frontline antibiotics. The ESKAPE bacteria (*Enterococcus faecium*, *Staphylococcus aureus*, *Klebsiella pneumoniae*, *Acinetobacter baumannii*, *Pseudomonas aeruginosa* and *Enterobacter* spp) have been highlighted by the World Health Organisation as the primary threats to human health associated with antibiotic resistance and acquisition of multi-drug resistance appears to be intrinsic to their success [2]. Opportunistic pathogens resistant to all first-line antibiotics are emerging in hospital intensive care units [3], sometimes accompanied by emergent mechanisms of drug resistance such as New Delhi metallo-β-lactamase [4] and very high levels of aminoglycoside resistance due to methylation of bacterial 16S rRNA [5]. Globally, nosocomial spread of such pathogens has sharply increased and is compounded by travel to endemic areas, facilitating importation of MDR bacteria into far-removed communities and environments [6]. The situation is exacerbated by a marked decline in the number of new antibiotics entering the marketplace and new therapies are badly needed to extend treatment options for life-threatening infections due to both Gram-positive and Gram-negative bacterial pathogens [7,8].

The majority of therapeutic agents in clinical use today have been sourced from naturally occurring agents and materials. Natural products and natural product structures continue to play a highly significant role in the drug discovery and development process. Nature’s chemical and biological diversity has been fruitfully exploited with regard to therapeutics that target bacterial infections; approximately 75% of antibacterial New Chemical Entities introduced into the clinic over the last forty years have been derived from natural products [9,10], without exception from moulds and actinomycetes, notably *Streptomyces* spp [11]. Reductions in the rate of discovery of novel molecules, drug scaffolds and antibacterial pharmacophores have increased the difficulty and the cost of identifying novel antibiotics by traditional methods. The decline has coincided with massive investment by the pharmaceutical industry into target-based drug discovery and reliance on emerging molecular technologies such as whole-genome sequencing and robotic screening of very large combinatorial libraries in order to provide an abundance of new targets and to enhance the capacity to identify new agents within existing chemical diversity [12,13]. For a variety of reasons [12,14], these hopes have not been realised.

Clearly, it is opportune to consider alternative sources and novel treatment paradigms for the generation of new treatments for bacterial infections, in particular for those caused by MDR pathogens, and this is increasingly fertile ground for infectious disease researchers [7]. Much of the global population depends on traditional, often plant-based, medicines for their health and well-being and there is a perception, frequently unfounded, that such treatments often have a positive outcome [15]. The rise of alternative medicine is in part driven by an often invalid belief in the efficacy and safety in natural products and this faith is easy to exploit commercially [16]. There is a large body of data describing the in vitro activity of plant extracts and constituents, in part due to the low barrier for entry into this field by underfunded laboratories. The majority of these studies have utilised unfractionated extracts with weak antibacterial activity as determined by the minimum inhibitory concentration (MIC) against common pathogens such as *K. pneumoniae*, *P. aeruginosa* and *S. aureus* [7]. The precise mechanism of action of the large majority of such phytochemicals is unknown, and many resemble weak antiseptics and are unlikely to act through a target-specific mechanism in “lock-and-key” fashion [17], in all likelihood restricting their use to topical infections. Commercial screening of plant materials for potent, non-toxic, broad-spectrum antibiotics failed to find suitable lead structures or development candidates [17]. Nevertheless, it is clear that a minority of phytochemical compounds have the capacity to non-lethally modify the properties of major pathogens in a way that could be exploited from a pharmaceutical perspective; green tea-derived catechin gallates fall into this category and will be the focus of this review.

## 2. Antibacterial Activity of Catechins and Catechin Gallates

Although the antibacterial properties of black tea were demonstrated over one hundred years ago and its consumption was recommended by the British Army Medical Corps during the First World War to ward off infection [18], systematic research of the antimicrobial activities of tea and its major constituents did not begin until the late 1980s, with the focus on the bioactive constituents of green tea. These studies have shown that tea can inhibit and kill some but not all Gram-positive and Gram-negative bacterial species at or slightly below the typical concentrations found in brewed tea [19]; activity is attributable to the polyphenolic catechins, in particular (−)-epigallocatechin (EGC), (−)-epigallocatechin gallate (EGCg) and (−)-epicatechin gallate (ECg) [20,21], and is generally modest. A key focus for such work has been the highly successful opportunistic pathogen *S. aureus*, a leading cause of hospital- and community-acquired infections worldwide. Clones of this Gram-positive bacterium have accumulated genes conferring resistance to a wide variety of frontline antibiotics, typified by methicillin-resistant forms (MRSA) that are resistant to all β-lactam antibiotics [22]. MICs of a variety of naturally occurring and semi-synthetic catechins and catechin gallates against epidemic clones MRSA-15 and EMRSA-16 ranged from 64 to >256 mg/L [23]; these values define weak activity that precludes these compounds from systemic use as conventional antibacterial agents. Non-galloylated catechins displayed no direct antibacterial activity in these assays although bactericidal action could be markedly improved by substitution of the gallate moiety of ECg with 3-*O*-acyl chains of various lengths [24]. Antibacterial effects have also been shown against other Gram-positive bacteria, including *Bacillus anthracis* [25], *Bacillus cereus* [26], *Bacillus subtilis* [27] and *Clostridium botulinum* [28]. The abundant catechins tend to have less activity against Gram-negative, compared to Gram-positive bacteria, almost certainly due to the added complexity and heightened barrier function of the cell envelopes of the former. Moderate to very poor activity against the gastric pathogen *Helicobacter pylori* [29], some rough “laboratory” strains [30,31] and several clinical isolates [32] of *Escherichia coli*, *Pseudomonas aeruginosa* [32] and a range of Gram-negative phytopathogens [33] has been demonstrated.

There is a large body of anecdotal, and increasingly evidence-based, material suggesting that moderate consumption of green tea has a range of health benefits [34]. There is convincing evidence that the antibacterial components of green tea are able to protect to some extent against diseases of the oral cavity that have a microbial aetiology, namely dental caries, gingivitis and periodontitis, as well as oral malodour. Caries is the most prevalent global health condition [35]; it is a complex infection generally caused by the α-haemolytic streptococci *Streptococcus mutans* and *Streptococcus sobrinus* and involves dietary, nutritional, microbiological and immunological factors that come into play to create the necessary conditions to allow these bacteria to gain a foothold and cause the disease. Cariogenic bacteria initially adhere to the tooth surface and produce a sticky glycocalyx that may develop into a mixed biofilm that manifests as dental plaque; bacteria in the biofilm then produce acid from dietary carbohydrates that erodes the tooth enamel [36]. Catechins have been described as moderate inhibitors of the cariogenic bacterial agents at concentrations comparable to those found in brewed tea [37,38,39], although these direct effects have been disputed [40], and they prevent the attachment of oral streptococci, including *S. mutans*, to surfaces [41,42]. Production of the adherent extracellular polysaccharide glycocalyx is an essential virulence factor for enabling *S. mutans* to produce caries [43,44]; their formation is catalysed by secreted glucosyltransferases with sucrose as substrate and they promote the accumulation of cariogenic streptococci on the tooth surface, contributing to the structural integrity of dental plaque. EGCg and ECg, as well as the catechin fraction of tea, inhibit streptococcal glucosyltransferase [45,46,47]. Refined catechins [48] and tea extracts [49] reduce α-amylase activity in human saliva and therefore prevent the breakdown of starch from foods with cariogenic potential. ECg and EGCg inhibit up-regulation of interleukin-8 and prostaglandin E2 in explants of dental pulp tissue [40] and thus have the potential to dampen the inflammatory events that are critical for the establishment of caries. These observations underpin the results of field trials that have demonstrated the capacity of green tea to reduce the incidence of caries. Thus, Magalhães and co-workers [50] demonstrated that a green tea extract mouthwash provided protection from tooth dentine erosion and plaque index was significantly reduced by use of green tea rinses, mouthwashes and tea polyphenol tablets [51,52,53].

Periodontal disease describes a variety of chronic inflammatory conditions of the soft tissue surrounding the teeth, bone and ligament supporting the teeth [54]. The disease begins with localized inflammation of the gingiva initiated by bacteria in the dental plaque to establish a gingival pocket between the gums and the tooth margin in which reside large numbers of predominantly Gram-negative anaerobic bacteria such as *Porphyromonas gingivalis*, *Aggregatibacter actinomycetemcomitans* and *Prevotella intermedia* [55,56,57]. The inflamed gum bleeds readily and recedes; disease may lead to a chronic condition characterized by absorption of ligaments and weakening of bone, causing the teeth to loosen. Periodontal disease may contribute to the body’s overall inflammatory burden. In similar fashion to caries, a four-week regimen of mouth washing with a dilute catechin solution was found to alleviate some symptoms of the condition, including halitosis [58,59]. High concentrations of catechin gallates, especially EGCg, are able to inhibit the growth of some periodontal bacteria [60,61], leading to significant reductions in markers of gingivitis. Further, bacterial toxic end metabolites, protein tyrosine phosphatase and gingipains associated with periodontal disease are neutralized by EGCg [62,63,64].

More details of other human intervention trials with tea extracts and catechins can be found in the reviews by Taylor et al. [21] and Narotzki and co-workers [65].

## 3. Non-Lethal Effects on Bacterial Cells

Catechins, in particular catechin gallates, have the capacity to interact with bacteria and their extracellular products in a way that may be of benefit to mankind without necessarily causing lethal damage [21,66]. They have an affinity for lipid-containing structures at or near the bacterial surface [67,68] and can also bind to bacterial surface proteins [69] and secreted proteins such as extracellular toxins [70]. For example, Nakayama and colleagues [69] used transmission electron microscopy (TEM) to record selective deposition of EGCg onto the outer membrane-located porin protein OmpG of *E. coli* NBRC 3972 due to exposed Lys, Arg, and His basic amino acids which strongly interacted with the catechin gallate. Simulations suggested that EGCg entered the porin channel and bound to Arg residues present on the inner surface of the pore channel through hydrogen bonding, resulting in inhibition of the porin function. Such insights could be useful for the generation of novel therapeutic strategies. A group at LeHigh University, Bethlehem, PA have recently shown [70] that galloylated catechins have the capacity to inhibit the activity of the LtxA leukotoxin secreted by the oral pathogen *A. actinomycetemcomitans*. Pre-incubation of the toxin with the catechins increased the inhibitory action, indicating that they act on the protein rather than on the bacterium. The secondary structure of the toxin was dramatically altered in the presence of catechin gallates and resulted in inhibition of toxin binding to cholesterol, an important initial step in the cytotoxic mechanism. EGCg also reduces secretion of the toxin due to EGCg-mediated enhancement of LtxA affinity for the bacterial cell surface [71]. EGCg supressed the proliferation of vegetative *B. cereus* and *B. subtilis* but not the germination of spores of these bacteria [72]. TEM indicated that EGCg was present on the surface of vegetative cells, but no deposition was detected on spore surfaces, suggesting that the compound was not adsorbed by spores. These data have implications for food and beverage preservation. EGCg also reduced the virulence properties of the important enterohaemorrhagic pathogen *E. coli* O157:H7 by attenuating the cross-species bacterial communication signal autoinducer 2 at non-inhibitory concentrations, thereby decreasing biofilm formation and swarm motility and increasing host survival in a *Caenorhabditis elegans* infection model [73].

The most intensively studied aspect of the impact of catechins on bacteria has been the capacity to reverse the resistance of the ESKAPE pathogen *S. aureus* to β-lactam antibiotics at concentrations well below the MIC [20,21,74,75]. The compounds are also able to modify the virulence properties of MRSA, including prevention of the formation of biofilms. Yam and colleagues [20] noted that extracts of green tea have the capacity to reverse methicillin resistance in MRSA clinical isolates; examination of individual tea constituents indicated that abrogation of reversible methicillin resistance was attributable to the catechin gallates, with ECg showing greater potency than either EGCg or Cg [76]. Others also demonstrated that ECg [74] and EGCg [75] markedly lower the MIC of methicillin, oxacillin and other β-lactam antibiotics in clinical isolates of MRSA. Catechins lacking the gallate moiety had no capacity to reverse methicillin resistance [23]. The effect was evident with all β-lactam agents examined (methicillin, oxacillin, flucloxacillin, cefotaxime, cefepime, imipenem, meropenem) and against all forty strains from an international collection of MRSA clinical isolates [23]. Thus, moderate concentrations (3 mg/L or greater) of ECg, EGCg and Cg are able to reduce the MICs of β-lactams for MRSA strains from full resistance (usually >256 mg/L) to below the antibiotic breakpoint (ca. 1–2 mg/L) [20,23], raising the possibility that these molecules could be used in combination with suitable β-lactam agents to treat MRSA infections.

These early studies indicated that the galloyl group (D-ring; Figure 1) is essential for β-lactam resistance-modifying activity and epi (*cis*)-configured catechin gallates, such as ECg, are more active than their *trans* counterparts, such as Cg, indicating that the stereochemistry of the C-ring partly governs bioactivity. The relative activities of catechin gallates suggest that a reduction in hydroxylation of the B-ring increases resistance-modifying capacity: for example, EGCg is less active than ECg although it differs from this molecule only by the presence of an additional hydroxyl group on the B-ring. Naturally occurring catechin gallates are variably absorbed from the intestinal tract and are rapidly metabolized to inactive products in this compartment due in part to the presence of ester bonds that are susceptible to hydrolysis by bacterial esterases [77]. These features were considered in order to design, synthesise and evaluate ECg derivatives with a more appropriate pharmacological profile for administration to man. The corresponding galloyl amide was synthesised and unlike the natural product was refractory to esterase degradation; it maintained the epi (*cis*) stereochemistry of ECg and was as active as the parent molecule in β-lactam modulation assays [78]. In addition, B-ring-modified derivatives of ECg, such as monohydroxyl compounds and a 3,5-dihydroxy B-ring catechin gallate, were synthesised and were at least as bioactive as ECg [79,80]. These studies suggest that the biological profiles of catechins, in particular galloylated compounds, may be improved by structural modifications.

## 4. Interactions of Galloylated Catechins with MRSA

Catechins exert their main biological effects through intercalation into lipid bilayers [67,82,83] and this in part is likely to explain the structural requirements for β-lactam resistance modification. Catechin gallates bind more avidly to small unilamellar phosphatidyl choline (PC) vesicles than (−)-epicatechin (EC) or (−)-epigallocatechin (EGC), the non-galloyl counterparts to ECg and EGCg, and they penetrate deep within the hydrophobic core of the lipid palisade [67]. ECg had a greater affinity for the bilayer than EGCg [84]. The relative affinity of catechins for membranes reflects their partition coefficients in *n*-octanol-saline [84] and their capacity to modulate β-lactam resistance [24]. Partition into PC liposomes paralleled their β-lactam resistance-modifying capacity, as catechins with epi (*cis*) stereochemistry intercalated more readily than their non-epi counterparts [82]. With epicatechins such as ECg, the hydrophobic domain in the region of the ester bond and C-ring (Figure 1) is relatively exposed and therefore more liable to perturb the bilayer. Reducing or modifying B-ring hydroxylation increases the dimensions of this hydrophobic domain and modulates the capacity of such unnatural catechins to partition into the phospholipid palisade [81].

These membrane-binding patterns were maintained when intercalation of catechins into lipid vesicles modelled on the staphylococcal bilayer was examined [81]. Unusually, the staphylococcal cytoplasmic membrane is composed of a complex asymmetric mixture of lipids with differing charge characteristics, predominantly positively charged lysylphosphatidylglycerol (LPG) and negatively charged phosphatidylglycerol (PG) and cardiolipin (CL) [68,85,86]. Incorporation of ECg into LPG:PG:CL synthetic membranes (natural lipid ratios of 53:43.5:3.5) produced significant changes to the thermotropic and anisotropic properties of the bilayer, with broadening of the phase transition peak indicative of the presence of phospholipid domains with strong hydrophobic interactions. EC did not induce significant changes to the calorimetric profile of the bilayer but could modify the ECg phase transition peak when the two molecules were simultaneously present within the lipid palisade [81], producing a significant reduction in vesicular membrane fluidity and a high degree of non-lethal membrane perturbation. These observations complement studies of catechin binding to MRSA cells: cell binding of ECg was enhanced by the presence of EC and EC and EGC significantly increased the capacity of ECg and EGCg to reduce levels of staphylococcal β-lactam resistance, even though these non-galloylated catechins had no impact by themselves on antibiotic susceptibility [87].

The insertion of ECg or EGCg into the staphylococcal cytoplasmic membrane [68,88] produces a complex phenotype, as could be anticipated from perturbation of such a key cellular constituent. Both compounds prevent and inhibit biofilm formation [89,90], a necessary prelude to staphylococcal invasion [91], disrupt the secretion of virulence-related proteins [92], compromise osmosensing in halotolerant staphylococci by interfering with Na^+^-specific antiporter systems [93] and promote cell wall thickening and cell aggregation without affecting the rate or extent of bacterial growth, indicating poor separation of daughter cells following division [90], in addition to their substantial impact on β-lactam resistance.

In recent years, much light has been shed on the mechanism(s) whereby galloylated catechins compromise resistance to second-generation β-lactams such as methicillin and oxacillin. The penicillin-binding proteins (PBPs), associated with the external surface of the CM and the cell wall, constitute the transglycosylases and transpeptidases that catalyse the insertion and cross-linking of newly synthesised peptidoglycan precursors in to the wall of Gram-positive bacteria such as *S. aureus* and are the targets for β-lactam antibiotics [94]. β-lactam resistance is associated with an additional PBP, PBP2a, which has a lower affinity for β-lactam antibiotics, functions cooperatively with PBP2 in the presence of high concentrations of β-lactams but displays poor transpeptidase activity in the presence of β-lactam concentrations that saturate other PBPs. Thus, MRSA continues to cross-link peptidoglycan chains in the presence of β-lactams even though the transpeptidase domain of PBP2 is inactivated [95]. Synthesis of the pentaglycine cross-bridge, used by PBPs to cross-link glycan chains in peptidoglycan, is catalysed by the enzymes FmhB, FemA and FemB using tRNA_gly_ as glycine donor [96,97]. They are located on the inner surface of the staphylococcal cytoplasmic membrane and their substrates are physically linked to the membrane [96,98]. Loss of FemA and/or FemB activity results in cells that are hypersusceptible to β-lactams, as PBP2a cannot accept the resulting tri- or mono-glycine interpeptides as substrates [98]. The expression of genes coding for these proteins was not modulated by ECg [68]. Growth of MRSA in medium containing β-lactam-modulating concentrations of ECg did not impact on transcription of the *mecA* gene encoding PBP2a, or on the cellular or cytoplasmic membrane concentrations of PBP2a protein, and there is no evidence that ECg compromises β-lactam resistance by binding directly to peptidoglycan [90]. ECg produced only small changes in the expression and functionality of other PBPs, resulting in a 5–10% reduction in peptidoglycan cross-linking without compromising cell integrity [90]. Such studies imply a complex basis for β-lactam resistance modification by galloylated catechins in MRSA clinical isolates.

Continuous peptidoglycan synthesis is critical for the maintenance of staphylococcal cell integrity and viability and for the facilitation of cell replication. Newly synthesised peptidoglycan components are incorporated into the intact cell wall after cleavage by autolysins (peptidoglycan hydrolases) in a dynamic network requiring an ordered balance between synthesis and degradation of peptidoglycan during bacterial growth. Autolysin-regulated orderly cell division, cell wall turnover and cell separation of Gram-positive bacteria are known to be partly dependent on the charge characteristics of the cell wall and autolysins play an important role in the lysis of bacteria induced by β-lactam antibiotics [99]. Secretion of autolysins during the cell division cycle is a prerequisite for cell separation [100]. ECg-mediated perturbation of cell division, cell wall turnover and cell separation are likely to be due to changes in the secretion of autolysins from the cell: ECg-grown cells retained autolysins within the thickened cell wall in a predominantly inactive form, with greatly reduced amounts released into the growth medium [90], and this almost certainly accounts for the reduction in cell separation observed by a number of groups [90,101]. The processing or enzymatic activity of the autolysins in the cell wall may be compromised, perhaps through binding to charged components [99], as ECg induced a large reduction in Triton-X-100-induced autolysis [90]. Thus, the abnormal wall thickness of ECg-grown cells is probably due to lower rates of cell wall degradation rather than increased rates of cell wall biosynthesis.

Activity and retention of autolysins are also influenced by the distribution of charge associated with the cell wall [99]; the net positive charge of the cell surface is reduced during growth in the presence of ECg (Figure 2) due to a reduction in the degree of d-alanylation of peptidoglycan-associated wall teichoic acids (WTAs) [102], leading to electrostatic repulsion from negatively charged surfaces and a failure to develop as a biofilm [99]. These poly-d-ribitol-phosphate wall components are known to have a significant impact on other properties of the Gram-positive cell wall, including modulation of autolysin activity, cation homeostasis, acid tolerance, expression of virulence determinants and susceptibility to β-lactam and other antibiotics [99]. The d-Ala content of WTAs from MRSA isolates BB568 and EMRSA-16 was reduced substantially by 50% and 43%, respectively, and molecular simulations indicated a decrease in the positive charge of the cell wall, confirmed by increased binding of cationised ferritin (Figure 2) and an increase in WTA chain flexibility to a random coil conformation [102]. ECg also engendered release from MRSA BB568 of the other form of teichoic acid associated with the staphylococcal cell wall, lipoteichoic acid (LTA), presumably displaced from the phospholipid palisade by the intercalating molecules [90]. These changes emphasise the huge impact that galloylated catechins exert on staphylococci and explain some aspects of the complex ECg-induced MRSA phenotype, although the profound alterations in β-lactam susceptibility engendered by ECg are likely to be due to direct interference with the antibiotic resistance machinery of the cell.

Further evidence that galloyl catechins interact predominantly with the staphylococcal cytoplasmic membrane has come from studies of the impact of exposure to ECg on the gene expression of EMRSA-16 [68,81]. Significantly up-regulated genes were involved in cation transport and binding, transport and metabolism of amino acids and riboflavin, electron transport, β-lactamase expression, shape determination and pathogenesis but most prominent were those belonging to the general cell wall stress stimulon. These genes are induced following exposure to cell wall-active antibiotics such as vancomycin, oxacillin, d-cycloserine, bacitracin and daptomycin and to cationic defence peptides [103,104,105]. Thus, EMRSA-16 cells respond to ECg by taking steps to preserve and repair a compromised cell wall and membrane. EMRSA-16 genes down-regulated by ECg included those encoding osmotic shock proteins, membrane proteins and exported proteins and lipoproteins, genes involved in nitrate-nitrite, amino acid and carbohydrate metabolism, and protein and ion transport. In particular, genes of the *agr* regulon, controlling the expression of proteins involved in pathogenesis [106], were strongly down-regulated and provide a firm basis on which to explain ECg-induced reductions in the expression of virulence determinants [92]. Bernal et al. [68] also determined sites of [^3^H]ECg binding to EMRSA-16: labelled ECg rapidly entered the cytoplasmic membrane to produce an immediate reduction in bilayer fluidity but cells rapidly responded by altering the composition of the membrane, incorporating a greater complement of branched chain fatty acids to re-establish a fluid phospholipid palisade. A proportion (38%) of [^3^H]ECg was recovered from the cell wall fraction, additional to the membrane (58%), suggesting alternative binding site(s) that may contribute to the altered phenotype. Only a small quantity (~10%) of radiolabel was found in the intracellular fraction, raising doubts about the relevance of studies showing inhibition by galloylated catechins of intracellular enzymes such as fatty acid synthetases and DNA gyrase in cell-free systems [107,108]. The pronounced asymmetry with respect to the distribution of phospholipids between the inner and outer leaflets of the EMRSA-16 bilayer was found not to be radically altered [109] but ECg intercalation reduced the content of LPG in the bilayer, with concomitant increase in the amount of PG [68]. Membrane-anchored MprF attaches lysine residues to PG and the resulting LPG is translocated to the outer leaflet of the membrane, where its positive charge profile provides protection against positively charged antibacterial peptides and daptomycin [110]. ECg intercalation compromises MprF [68], as reflected in up-regulation of *mprF* following exposure to the galloylated catechin [81]. Deletion of *mprF* from EMRSA-16 and a DAP-resistant mutant derived from this clinical isolate substantially increased net negative surface charge but had no effect on β-lactam resistance in the presence or absence of ECg [81]; the *mprF* gene product is, therefore, unlikely to contribute towards ECg-determined increases in β-lactam susceptibility.

PBP2 is normally located at the division septum in *S. aureus* [111], the primary locus of cell wall synthesis, and is recruited to the division site by externalized peptidoglycan precursor substrate molecules [112]. β-lactam antibiotics bind covalently to the transpeptidase active site of PBP2 and abrogate catalytic activity, preventing formation of pentaglycine cross-bridges and, in β-lactam-susceptible strains, delocalising PBP2 from the septum. As PBP2 and PBP2a function cooperatively in the presence of oxacillin to ensure orderly peptidoglycan synthesis at the septum, there is no delocalisation of PBP2 in MRSA strains [112]. Cell division and recruitment of proteins of the cell replication machinery, the divisome (Figure 3), are initiated by polymerisation into a filamentous Z-ring structure of the cytosolic tubulin homologue FtsZ, a self-activating guanosine triphosphatase [113,114]; the divisome contains the enzymes that catalyse addition of new monomers to the peptidoglycan envelope [115]. FtsZ remains at the septum after exposure of MRSA to ECg but ECg intercalation into the bilayer delocalises PBP2 from the septum [68] and physically and functionally decouples PBP2 and PBP2a [116]. It is highly likely that this uncoupling, where PBP2a is no longer able to compensate for the loss of PBP2 transpeptidase activity in the presence of β-lactam antibiotics, fully accounts for the β-lactam-sensitising capacity of galloylated catechins. This mode of action raises the possibility that combinations of β-lactam agents and divisome-disrupting cytoplasmic membrane intercalators could be employed to restore the efficacy of β-lactam antibiotics whose utility has been eroded by the emergence of drug resistance. A role for catechin gallates in the treatment of MRSA infections has been extensively discussed in a recent review [88].

## 5. Concluding Remarks

The benefits of tea drinking have been known since antiquity and the perception that tea consumption, particularly of green tea, contributes to well-being and longevity have been bolstered by studies using modern scientific methods. The bioactive components of green tea, the catechins and catechin gallates, have been intensively investigated for their antioxidant, anti-inflammatory and metabolic regulatory activities, and may have significant impact on the common lifestyle diseases that continue to bedevil societies in both Eastern and Western cultures. It is becoming clear than many of the diverse and potentially exploitable effects of these compounds are due to interactions with both prokaryotic and eukaryotic cells at the cell membrane level. The large majority of plant-derived molecules, even those that have evolved to protect plants against bacterial and fungal infection, are unsuitable for development as anti-infective agents for use in man and animals, as plants exploit different mechanisms for defence against microbes. Catechins are no exception; their MICs against important bacterial pathogens are generally very weak and effective only at concentrations that are unobtainable in vivo; but their capacity to fully reverse β-lactam resistance in MRSA at concentrations an order of magnitude below the MIC is impressive and could point the way to a new generation of phenotypic modifiers that resolve infections due not to their antibacterial properties reflected in the MIC but to their ability to disperse drug resistance machineries. Catechins are rapidly metabolised once they are introduced into mammalian subjects by the common routes, so they themselves are unlikely to be useful without substantial chemical modification. We attempted to engineer these chemically complex molecules to achieve more acceptable compounds from a pharmaceutical perspective; although improvements can be made, the tortuous chemistry involved confounded attempts at scale-up. However, the principle of selective membrane perturbation, as illustrated by the interaction of catechins with the staphylococcal cytoplasmic membrane, would seem a valid and novel approach to the development of novel antibacterial agents.

## Figures and Tables

**Figure 1 molecules-25-01986-f001:**
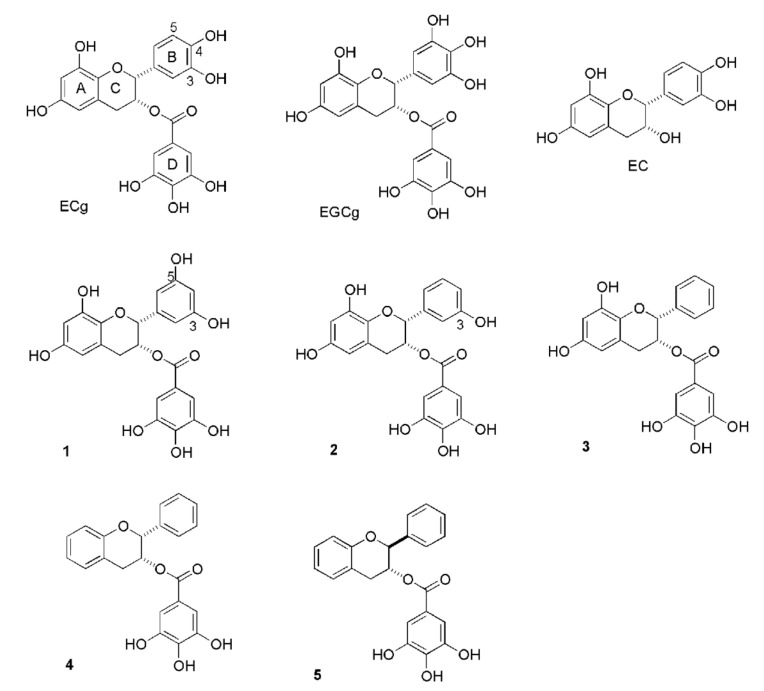
Structures of (−)-epicatechin gallate (ECg), (−)-epigallocatechin gallate (EGCg), (−)-epicatechin (EC), (−)-3,5-dihydroxy B-ring modified (−)-ECg (1), (−)-3-hydroxy B-ring modified (−)-ECg (2), (−) B-ring modified (−)-ECg (3), (−)-A,B-ring modified (−)-ECg (4) and A,B-ring modified (−)-Cg (5). Reproduced from Palacios et al. [81] under the terms of the Creative Commons Attribution license.

**Figure 2 molecules-25-01986-f002:**
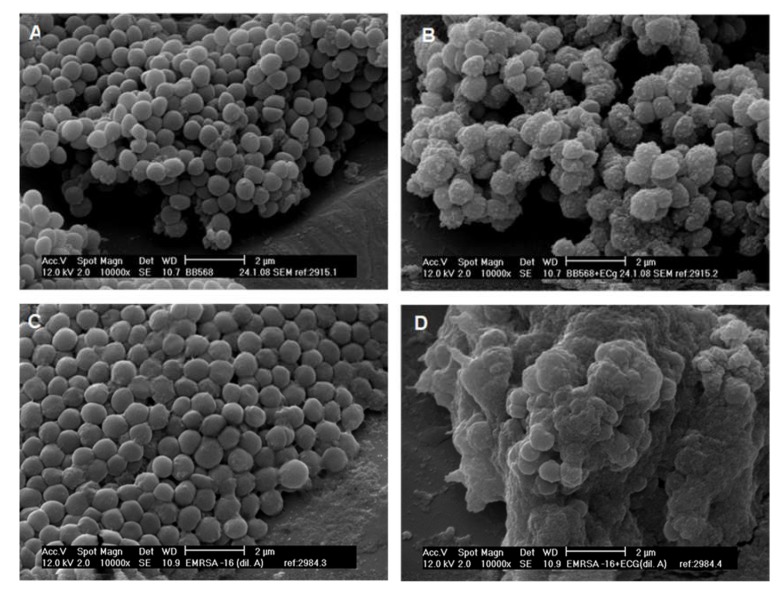
Scanning electron micrographs of *S. aureus* BB568 cells grown in non-supplemented Müller–Hinton broth (MHB; **A**), in MHB containing 12.5 mg/L ECg (**B**). BB568 cells were incubated with cationised ferritin after growth in non-supplemented (**C**) and ECg-supplemented (**D**) MHB. Reproduced from [7] with permission from RightsLink/Elsevier.

**Figure 3 molecules-25-01986-f003:**
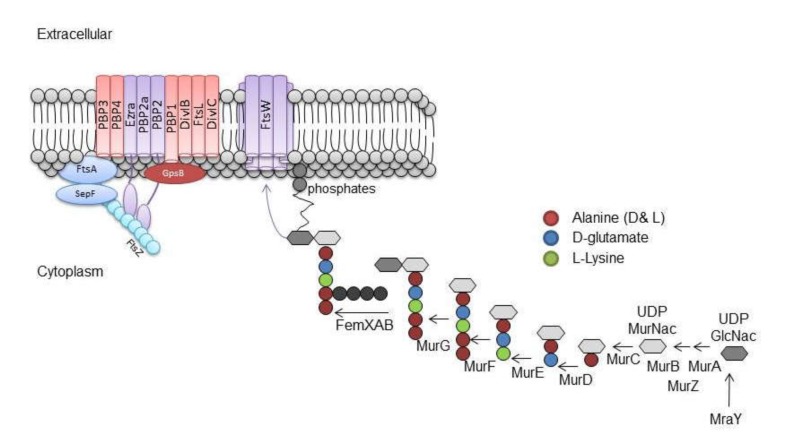
Proteins of the divisome recruited to the division septum of methicillin-resistant *Staphylococcus aureus* (MRSA) during cell division. Cytoplasmic synthesis of lipid II, the peptidoglycan precursor polymerised by penicillin-binding proteins (PBPs), is also shown in the not-to-scale representation. (Figure kindly provided by Dr. Sarah Paulin; from her PhD thesis 2014).

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
