# Peer review of "Interactions of Tea-Derived Catechin Gallates with Bacterial Pathogens"

_molecules, 2020, doi:10.3390/molecules25081986_

Round 1

Reviewer 1 Report

Dear author, congratulations by this work. However, in the text, I suggeste revise the references to standardiz the references (i.e. reference 14);

Other point, in your review, you ignored completely the work of Dr. Tintino and Morais-Braga, that works with natural products associated with antimicrobials to many years.

I  suggest for you seek and use some papers from these researchers to enhance the diversity of groups of your work, with different view points (Please seek on PUBMED by Tintino SR and Morais-Braga MF).

Author Response

The first point raised is to standardise the references, with ref 14 given as an example. I am not sure what is meant by this; I was careful to ensure that all references conform to the journal's format and ref 14 does this as far as I can see. If you are questioning the suitability of the reference, I point out that this is a good, comprehensive and up-to-date review of the failure of "genomic" drug discovery combined with the use of large combinatorial chemical libraries to identify new antibiotics within the confines of existing active chemical scaffolds and I believe this reference fits the bill very well.

The second point you raise concerns the fact that I do not cite the work of Tintino and Morais-Braga. I was invited to write the current review for the journal "Molecules" by Japanese scientists who investigate the impact of tea-derived catechin gallates on bacterial phenotypes, in particular MRSA clones. I know the relevant literature in this field particularly well as I have actively researched the interaction of MRSA  and these tea components for around twenty years. I had not come across the work of these two authors whilst researching this topic. I did as asked and undertook a PubMed search and obtained twelve hits from 2012 to 2020. None of these have anything to do with the topic I was asked to address and I am at a loss to understand the request that I include any of these references in my review. The majority of these studies are published in obscure journals (African Health Sciences, Scientifica Cairo, Saudi Journal of Biological Sciences) with really poor impact factors - for example, the International Journal of Vitamin and Nutrient Research has a current impact factor of 1.25. I have carefully read all twelve papers and the majority describe the use of crude, undefined plant extracts (ferns, seaweed, herbaceous plants, etc) in MIC assays, sometimes in combination with antibiotics such as gentamicin, with very unimpressive outcomes. There are a very large number of studies in this area and much of it is meaningless - any living organism is likely to contain compounds with weak activity against microorganisms and I see little point in the use of extracts containing undefined chemicals to produce MICs that have no clinical relevance whatsoever. I demolished such studies in a 2013 review I wrote by invitation for the International Journal of Antimicrobial Agents which I cite in the current review as ref 7. The majority of the twelve papers by these authors concern activity against fungi and parasites, a few against bacteria, none involve constituents of tea and some involve the use of vitamins (K3) and alpha-tocopherol. None of this is relevant to the topic I was given for the current review under consideration by "Molecules".

Reviewer 2 Report

The review Interactions of Tea-Derived Catechin Gallates with Bacterial Pathogens describes the possible modes of action and mechanisms that explain how catechin gallates induce antibiotic sensitivity in methicillin-resistant Staphylococcus aureus (MRSA) strains. The topic is of great interest and relevance for suggesting and divulging possible alternatives to combat antibiotic resistance. The review explains very clearly all the background of the topic and greatly facilitates the understanding of the mechanisms involved. The references include a very updated bibliography. The article shows a careful format that meets in my opinion the standards of the journal.

Author Response

I thank the reviewer for their positive comments. I welcomed the opportunity to review this field as there have been some significant recent advances and I strove to accommodate these in the article.